# A key indicator of nicotine dependence is associated with greater depression symptoms, after accounting for smoking behavior

Tiffany Bainter[1], Arielle S. Selya[1,2,3]*, S. Cristina Oancea[1]

1 Department of Population Health, Master of Public Health Program, University of North Dakota School of Medicine & Health Sciences, Grand Forks, ND, United States of America, 2 Behavioral Sciences Group, Sanford Research, Sioux Falls, SD, United States of America, 3 Department of Pediatrics, University of South Dakota Sanford School of Medicine, Sioux Falls, SD, United States of America

* arielle.selya@sanfordhealth.org

**Data Availability Statement:** The datasets analyzed during the current study are publicly and freely available on the NHANES website, https://www.cdc.gov/nchs/nhanes/.

## Abstract

### Introduction

Depression is a global burden that is exacerbated by smoking. The association between depression and chronic smoking is well-known; however, existing findings contain possible confounding between nicotine dependence (ND), a latent construct measuring addiction, and objective smoking *behavior*. The current study examines the possible *unique* role of ND in explaining depression, independently of smoking behavior.

### Methods

A nationally-representative sample of current adult daily smokers was drawn by pooling three independent, cross-sectional, biennial waves (spanning 2011–16) of the National Health and Nutrition Examination Survey (NHANES). The association between ND (operationally defined as time to first cigarette (TTFC) after waking) and the amount of depression symptoms was examined after adjusting for both current and lifetime smoking behaviors (cigarettes per day and years of smoking duration) and sociodemographic factors (gender, age, race, education and income to poverty ratio).

### Results

Earlier TTFC was associated with more depression symptoms, such that those smoking within 5 minutes of waking had an approximately 1.6-fold higher depression score ($PRR = 1.576$, 95% $CI = 1.324$–$1.687$) relative to those who smoke more than 1 hour after waking. This relationship remained significant after adjusting for current and lifetime smoking behavior as well as sociodemographic factors ($PRR = 1.370$, 95% $CI = 1.113, 1.687$).

**Funding:** This work was supported by the National Institute for General Medical Sciences (NIGMS, URL: https://www.nigms.nih.gov/) in the National Institutes of Health (NIH), through grant number P20GM121341 to PI Weimer, which supported AS's contribution. The funders had no role in study design, data collection and analysis, decision to publish, or preparation of the manuscript.

**Competing interests:** The authors have declared that no competing interests exist.

## Conclusions

The latent construct of ND, as assessed by TTFC, may be associated with an additional risk for depression symptoms, beyond that conveyed by smoking behavior alone. This finding can be used for more refined risk prediction for depression among smokers.

## Introduction

Among Americans, 8.1% of adults aged 20 and older reported depression in the last 2 weeks during years 2013–2016 [1]. Major depression imposes a high cost burden to society, and functional impairment resulting from depression is greater than other chronic diseases such as diabetes and arthritis [1]. Globally, depression is a substantial cause of disability, affecting over 300 million people [2].

Those with a mental health condition live on average 13 years less than those in the general population, with smoking-related diseases being the largest contributor to these early deaths [3]. The association between smoking and depression is well-known: those with depression are more likely to be smokers, smoke more heavily, and experience nicotine dependence [4], with a bi-directional temporal relationship between smoking and depression [5]. Current smokers have significantly higher depression rates compared to former-smokers and never-smokers [6, 7]. Importantly, among young adults, the increased risk of major depressive disorder (MDD) associated with smoking can be reduced by quitting [8].

In this body of literature, it is important to distinguish nicotine dependence (ND) from smoking *behavior*. While smoking behavior is an objective measure of tobacco *consumption*, ND is a latent construct which indicates the extent of psychological and/or physiological addiction to nicotine [9, 10, 11]. Traditional measures intended to capture smoking addiction typically capture objective smoking behavior (e.g. cigarettes per day, pack-years). However, these measures are markedly imperfect. For example, smoking behavior and ND are separable when comparing different subpopulations of smokers: "chippers" are a subgroup of smokers that do not show signs of dependence, despite extensive and regular smoking history [12]. Conversely, ND symptoms have been consistently reported among some adolescent smokers, even soon after initiation and at low levels of smoking [13–16]. That is, a given smoker can have heavy smoking behavior without ND, and vice versa. Further, traditional measures of smoking behavior (here, cigarettes per day) correlate poorly with biomarkers of smoking such as cotinine, a metabolic byproduct of nicotine [17], possibly due to widely varying smoking topographies (length of inhalation, number of puffs, etc.) across smokers [18, 19] which can result in drastically different levels of nicotine extracted per cigarette [20–24]. Thus, objective smoking behavior provides only limited information about later smoking patterns or smoking exposure.

ND, on the other hand, captures the underlying degree of addiction or dependence. A variety of psychometrically validated scales have been developed to measure ND, including the Fagerström Test for Nicotine Dependence (FTND) [25] and the Heaviness of Smoking Index (HSI) [26] which contain items on objective smoking behavior as well as psychological and/or physiological dimensions of ND such as urgency to restore nicotine levels after abstinence and persistence of nicotine levels during waking hours [27]. A criticism of some ND scales is their reliance on objective smoking behavior and more severe indicators of addiction [28], which some groups (e.g. adolescents) do not meet criteria for, but are nonetheless dependent according to other dimensions [13–16]. Other scales have been developed which do not include

objective behavioral items; for example, the Nicotine Dependence Syndrome Scale assesses five physiological and/or psychological dimensions: drive, priority, tolerance, continuity, and stereotypy, which can each be present without severe or heavy smoking behavior [29]. The item time to first cigarette (TTFC) which is on the FTND and the HIS, is of particular note. Although TTFC appears to be an objective smoking behavior, it is a robust and reliable indicator of overall nicotine dependence [9], possibly because it indicates physiological dependence [11, 17]. When measuring overall ND (and not focusing on individual dimensions of ND), TTFC is the best single-item indicator [9]. Thus, TTFC is a highly efficient indicator of overall ND.

This distinction between smoking behavior and ND is important with respect to a variety of behavioral and health outcomes. ND and smoking behavior have statistically independent and additive relationships with later smoking behavior [13, 14, 30], as well as with smoking-related health outcomes. For example, even after accounting for smoking behaviors, ND remains additionally and significantly associated with a higher risk for lung, larynx, head and neck cancer [31–33], more severe pulmonary impairment [20], less favorable cholesterol profiles [21], and a higher risk for asthma [22].

Several studies have made this important distinction between ND and smoking behavior in examining the relation to depression. Those with depression have higher levels of nicotine dependence [4, 34–39], and vice versa [34, 40]. However, very few studies *controlled for* smoking behavior when examining the association between depression ND [34, 4, 39], meaning that it remains unclear what the unique contributions of ND and smoking behavior are to explaining depression outcomes. Longitudinal research has found that major depression is a predictor of earlier TTFC, controlling for a limited measure of current smoking behavior [36] among Canadian adult smokers. Very little is known about the unique contribution of ND in explaining depression, over and above detailed measures of smoking behavior (including smoking history), among a nationally-representative US sample. If ND proves to explain depression more strongly than objective smoking behavior, this is highly relevant for depression screenings among smokers. Previous research has established precedence that ND is a better predictor of other outcomes (e.g. cessation outcomes and smoking biomarkers) when compared to the fairly weak explanatory power of cigarettes per day [11, 17].

The current study examines the distinct relationship between TTFC a robust and reliable indicator of nicotine dependence [9], and outcomes of depression symptoms (which assesses both the number and severity of individual MDD symptoms) among a nationally-representative sample of current adult smokers in the US, drawn from the National Health and Nutrition Examination Survey (NHANES), waves 2011–12, 2013–14, and 2015–16. Weighted regressions will examine the *independent* association between TTFC and depression symptoms after controlling for current and lifetime smoking behavior as well as sociodemographic risk factors for smoking.

## Materials and methods

### Ethics statement

Ethical approval for analysis of these existing, publicly available data was given by the University of North Dakota IRB on 11/2017 under project number IRB-201708-021. Sanford Research deemed this study exempt from human subjects research review due to being an analysis of existing data.

### Sample

Data were drawn from the large, nationally-representative NHANES survey [26], pooled across years 2011–2012, 2013–2014 and 2015–2016, which are each cross-sectional and

contain independent samples. NHANES is representative of the civilian, noninstitutionalized population in the Unites States, with each biennial wave consisting of about 10,000 respondents. NHANES collects demographic, dietary, socioeconomic, and health-related questions by interview. NHANES also conducts examinations and laboratory tests for medical, dental and physiological data. The current study focused on $N$ = 2070 participants aged 20 years and over who reported current daily smoking, due to availability of data on smoking behavior. Data from the "Demographics" file and the questionnaires "Cigarette Use" and "Mental Health-Depression Screener" were used in this analysis.

## Availability of data and materials

The datasets analyzed during the current study are publicly and freely available on the NHANES website, https://www.cdc.gov/nchs/nhanes/.

## Measures

**Major Depressive Disorder (MDD).** Outcomes of depression were measured using responses from the NHANES questionnaire "Mental Health-Depression Screener" which was based on the 9-item Patient Health Questionnaire (PHQ-9) [41,42]. The PHQ-9 incorporates criteria from the Diagnostic and Statistical Manual, Fifth Edition (DSM-5) criteria for MDD [43,44], with 9 items assessing symptoms of depression within the last 2 weeks and a 10[th] item assessing the functional impairment due to these symptoms. Each item was assessed on a 4-point scale ranging from 0 (e.g. "not at all") to 3 (e.g. "nearly every day"), with larger numbers indicating more depression symptoms. The major outcome variable of depression symptoms was created by summing these 10 items, as recommended by NHANES [45], into a single variable with a possible range of 0–30, with higher numbers indicating more depression symptoms.

For preliminary analyses only, a binary outcome variable was created by coding these 10 items according to the DSM-5 criteria for major depressive disorder [43]. Specifically, the first 9 symptoms were coded as binary based on the symptom being present 'nearly every day' (1) vs. 'not at all', 'several days' or 'more than half the day' (0) consistent with the DSM-5 criteria. Next, a final classification of presence vs. absence of MDD criteria was derived based on 1) experiencing 5+ of the 9 symptoms, of which one must be either "depressed mood" or "loss of interest or pleasure," and 2) experiencing functional impairment due to these symptoms ('extremely difficult' vs. 'not at all', 'somewhat', or 'very difficult').

**Time To First Cigarette (TTFC).** TTFC was assessed in the "Smoking—Cigarette Use" questionnaire using the question "How soon after waking do you smoke?" Original response categories are given on a 4-point scale in 2011–2012 ('within 5 minutes', 'from 6–30 minutes', 'from more than 30 minutes to one hour', and 'more than one hour') but on a 7-point scale in years 2013–2014 and 2015–2016 (with 3 additional categories ranging from 'more than 1 to 2 hours' to 'more than 4 hours'). To harmonize the differing response categories, the last 3 responses were collapsed for years 2013–14 and 2015–16 to align the variable with the 2011–2012 version, such that the highest TTFC value was 'more than one hour.'

**Smoking history.** Smoking history was also derived from the "Smoking—Cigarette Use" questionnaire, using both current and lifetime measures of smoking behavior. Lifetime smoking history was defined as duration of smoking in years, which was calculated as age of respondent minus the age they reported first starting to smoke regularly. Current smoking behavior was measured as the average number of cigarettes smoked per day during the past 30 days.

**Sociodemographic variables.** Additional confounding variables used in this study included gender, age, race, education, and income to poverty ratio as indicated by previous

literature [6, 8, 46, 47]. The variable of income to poverty ratio is a measure of family or individual income divided by the poverty specified for that year. The full range was 0–5, and was top-coded by NHANES at 5 to maintain confidentiality.

### Statistical analyses

A preliminary logistic regression was used to determine the relationship of TTFC with the presence of MDD criteria. TTFC was not found to be significantly associated with the odds of meeting clinical MDD criteria. These results are not presented in detail due to non-significance and the low number of cases meeting MDD criteria.

The relationship between TTFC and depression symptoms was examined first using an unadjusted, survey-weighted, quasi-Poisson regression model in the statistical software R. A quasi-Poisson model was used due to the depression outcome being defined as a symptom count; additionally, this type of model accounts for overdispersion due to excess 0's in the outcome variable (indicating absence of depressive symptoms). A second model adjusted for both lifetime and current smoking behaviors (years of smoking duration and cigarettes per day, respectively), as well as socio-economic variables (gender, age, race, education, and income to poverty ratio). The strength of the association between TTFC and depression was estimated using prevalence rate ratios 95% confidence intervals and $p$-values.

## Results

### Descriptive statistics

Table 1 illustrates the summary statistics in the sample, grouped by TTFC category. Those who smoke within 5 minutes of waking are approximately twice as likely (4.5%) to meet criteria for MDD than all other groups (1.7–2.8%), and more likely to have less than a high school education (35.3% vs. 23.2–26.6%). Earlier TTFC also had a monotonic relationship with smoking more cigarettes per day (median = 20 vs. 12 vs. 10 vs. 6 for TTFC within 5 minutes, from 5–30 minutes, from 30–60 minutes, and more than 60 minutes, respectively). However, there was no consistent monotonic relationship between TTFC and smoking duration.

### Depression score and TTFC

The unadjusted model indicates that smoking within 5 minutes of waking is significantly associated with a higher depression score by a factor of 1.576, compared to those who smoked more than 1 hour after waking (prevalence rate ratio ($PRR$) = 1.576, 95% confidence interval ($CI$) = 1.324–1.876, $p < .001$). No significant differences in depression symptoms were found when comparing TTFC > 60 minutes (reference group) vs. TTFC between 5–30 minutes ($PRR = 1.173$, $CI = 0.96–1.44$, $p = .128$) or TTFC between 30–60 minutes ($PRR = 1.144$, $CI = 0.93–1.41$, $p = .213$).

Table 2 shows the association between TTFC and the depression score, after adjusting for current and lifetime smoking behaviors (average cigarettes per day and smoking duration), as well as socio-economic variables (gender, age, race, education, and income to poverty ratio). Participants who smoke within 5 minutes after waking up had a depression score that was significantly higher by a factor of 1.370 relative to those that smoked more than 1 hour after waking ($PRR = 1.370$, $CI = 1.13–1.687$, $p = 0.005$) after controlling for current and lifetime smoking behaviors and socio-economic confounders. Those who smoked between 5 and 60 minutes after waking did not significantly differ in their depression scores than those who waited more than 1 hour after smoking.

**Table 1. Descriptive statistics by each level of Time To First Cigarette (TTFC) after waking.**

| Measure | | TTFC ≤ 5 min (N = 578) | TTFC 5–30 min (N = 685) | TTFC 30–60 min (N = 422) | TTFC >60min (N = 385) |
|---|---|---|---|---|---|
| Depression | Presence | 4.5% (N = 26) | 1.7% (N = 12) | 2.1% (N = 9) | 2.8% (N = 11) |
| | Symptom Count | 4 (1–10) | 3 (1–7) | 3 (1–7) | 2 (0–6) |
| Gender | Male | 57.1% (N = 330) | 55.9% (N = 383) | 55.7% (N = 235) | 58.2% (N = 224) |
| | Female | 42.9% (N = 248) | 44.3% (N = 302) | 44.3% (N = 187) | 41.8% (N = 161) |
| Age | | 46 (35–56) | 48 (34–58) | 44 (32–58) | 46 (31–60) |
| Race/ Ethnicity | White, Non-Hispanic | 55.0% (N = 318) | 55.6% (N = 381) | 47.2% (N = 199) | 34.0% (N = 131) |
| | Other-Hispanic | 10.2% (N = 59) | 9.6% (N = 66) | 14.2% (N = 60) | 25.2% (N = 97) |
| | Black, Non-Hispanic | 28.2% (N = 163) | 25.5% (N = 175) | 28.4% (N = 120) | 28.1% (N = 108) |
| | Other | 6.6% (N = 38) | 9.2% (N = 63) | 10.2% (N = 43) | 12.7% (N = 49) |
| Education | Less than 11th Grade | 35.3% (N = 204) | 26.6% (N = 182) | 23.2% (N = 98) | 26.2% (N = 101) |
| | High School or equivalent | 31.3% (N = 181) | 31.0% (N = 212) | 32.0% (N = 135) | 27.8% (N = 107) |
| | Some College, College Degree or above | 33.4% (N = 193) | 42.5% (N = 291) | 44.8% (N = 189) | 46.0% (N = 177) |
| Income to Poverty Ratio | | 1.08 (0.67–1.88) | 1.23 (0.77–1.71) | 1.65 (0.9–2.84) | 1.55 (0.88–2.78) |
| Cigarettes per Day | | 20 (10–20) | 12 (10–20) | 10 (6–15) | 6 (4–10) |
| Smoking Duration (Years) | | 29 (18–40) | 30 (16–41) | 24 (14–41) | 26 (12–39) |

Categorical variables are summarized as valid percentage (N) and quantitative variables are summarized as median (interquartile range).

Other significant findings from this model included significantly more depression symptoms among females ($PRR = 1.599$, $CI = 1.424$–$1.795$, $p < 0.001$) and the inverse relationship between income-to-poverty ratio and depression symptoms ($PRR = 0.873$, $CI = 0.829$–$0.918$, $p < 0.001$). Neither current nor lifetime smoking behavior was significantly associated with depression in the multivariate model.

**Table 2. Weighted poisson regression examining the association between TTFC and depression symptom counts.**

| Measure | | PRR | 95% CI | p-value |
|---|---|---|---|---|
| TTFC | ≤ 5 min. | **1.370** | **1.113–1.687** | **0.005** |
| | 5–30 min. | 1.088 | 0.875–1.353 | 0.452 |
| | 30–60 min. | 1.142 | 0.926–1.406 | 0.222 |
| | >60 min. | (Reference) | (Reference) | (Reference) |
| Cigarettes per Day | | 1.007 | 0.996–1.016 | 0.192 |
| Smoking Duration (Years) | | 1.005 | 0.995–1.015 | 0.346 |
| Gender | Male | (Reference) | (Reference) | (Reference) |
| | **Female** | **1.599** | **1.424–1.795** | **<0.001** |
| Age | | 0.994 | 0.983–1.004 | 0.260 |
| Race/Ethnicity | White, Non-Hispanic | (Reference) | (Reference) | (Reference) |
| | Other, Hispanic | 1.079 | 0.906–1.285 | 0.399 |
| | Black, Non-Hispanic | 0.939 | 0.829–1.062 | 0.324 |
| | Other | 1.160 | 0.919–1.463 | 0.219 |
| Education | Less than 11th grade | (Reference) | (Reference) | (Reference) |
| | High School/GED | 0.946 | 0.802–1.115 | 0.514 |
| | Some College, College Degree or above | 0.859 | 0.708–1.042 | 0.132 |
| **Income to Poverty Ratio** | | **0.873** | **0.829–0.918** | **<0.001** |

PRR (prevalence rate ratio), CI (confidence interval), TTFC (Time to First Cigarette). Min (minutes), hr/hrs (hour/hours). Bold: p < .05.

## Discussion

This study finds that after controlling for smoking behaviors and socioeconomic variables, there is an independent relationship between TTFC, a key indicator of ND, and the number of depression symptoms experienced. Specifically, those who smoke their first cigarette within 5 minutes of waking have on average a 1.37-fold higher count of depression symptoms relative to those who smoke their first cigarette more than one hour after waking.

The findings in this study confirm previous research characterizing the relationship between depressive symptoms and nicotine dependence [4, 6, 8, 13, 36, 47]. The current study extends this existing work by parsing out the statistically independent contributions of TTFC (a latent construct assessing *addiction*) versus objective smoking *behavior* and showing TTFC to have a unique, additional relationship with the number of depression symptoms. However, these findings only held true for depression *symptom count* but not for the *likelihood* of depression, in contrast with previous literature [6, 7]. This discrepancy could be due to the small number of participants who met the diagnostic criteria and/or the use of self-reported nature of depression symptoms, as opposed to clinical diagnoses by a trained professional. Further research using independent samples and longitudinal data is needed to further research the relationship between ND and depression outcomes.

A possible explanation of the current findings could be that smokers are able to titrate the nicotine extracted from each cigarette by altering their smoking topography (length of inhalation, number of puffs, etc.) [18,19]. More nicotine-dependent smokers may therefore extract more nicotine from each cigarette compared to non-dependent smokers with a similar smoking history [20–24]. The resulting higher exposure to nicotine may have direct physiological effects in the brain, consistent with previous research suggesting that TTFC measures physiological aspects of ND such as sensitivity to depletion of nicotine concentrations from overnight abstinence [11, 17, 48, 49]. Nicotine works similarly to other addictive drugs by activating reward pathways [50]. A salient symptom of nicotine dependence is withdrawal, which can include anxiety, restlessness, agitation and impaired concentration [50]. While the act of smoking can relieve these symptoms and give the feeling of calm and improve concentration, those with mental illness can interpret these effects as relieving the symptoms of their mental illness [50]. Thus, individuals with depressive symptomatology may smoke in order to self-medicate their mental health symptoms, but this may result in higher nicotine extraction. In turn, this exacerbates the addiction cycle by shortening the latency between cigarettes in order to relieve nicotine withdrawal symptoms [51].

Alternatively, nicotine addiction and depression may be associated through a common cause, consistent with prior research concluding that there is no direct causal relationship, but rather their association arises through another common factor [34] such as familial or genetic factors [5].

Our findings showed a non-monotonic relationship between TTFC and the number of depression symptoms, such that only those who smoke within 5 minutes of waking are different from those who smoke more than 1 hour after waking. The reasons for this are unclear, but one possibility is a threshold effect such that the association with depression symptoms only manifests at a certain level of ND. Threshold effects have been reported in other related research, e.g. the relationship between TTFC and cotinine levels [17]. Further research is needed to examine whether there is a continuous effect of TTFC on depression symptoms, or whether this represents a threshold effect.

Smoking can hinder mental illness treatment by limiting the amount of drug uptake used in many antipsychotics and antidepressants, which could result in higher dosing and possible medication toxicity if the individual attempts to quit smoking [50]. Smokers who are more

nicotine-dependent may smoke more heavily and more persistently [48], thus potentially compromising pharmaceutical mental health treatment and in turn exacerbating their depression symptoms. Taken together, this could explain the current and previous findings that indicators of ND are associated with more depression symptoms [4, 6, 8, 11, 47].

Researchers have found smoking cessation to be more complicated for those suffering from depression. Adults with major depression disorder have lower cessation rates and doubled relapse rates [47]. However, by quitting smoking, individuals may be at reduced risk for recurrence of mood or anxiety disorder compared to those that do not quit [52]. Taken together, smokers with depression stand to reap additional benefits by quitting smoking, but given the increased difficulty of doing so for this population, specialized cessation efforts are needed which are tailored to smokers suffering from depression [39]. Similar issues may exist for those experiencing depression symptoms that fall short of the threshold for a clinical diagnosis, as may be the case with the current sample.

## Strengths and limitations

A strength of this study the use of NHANES data which allows for a large, nationally-representative sample of adult smokers in the US. The sample size is greatly reduced due to the inclusion criteria of our study; however, the use of survey-weighted analysis indicates that these results are generalizable to the larger population of adult current smokers in the U.S. Nevertheless, the sample size is a limitation in that only 58 participants met DMS-5 criteria for binary major depression (preliminary results). Given the complexity of our fully-adjusted model, this low sample size may have contributed to the non-significant findings of this preliminary logistic regression.

Several limitations should be noted. First, NHANES collects data for depression symptoms by self-report, which may be biased. Second, these data are cross-sectional and thus the temporal relationship and causality between nicotine dependence and depression symptoms cannot be established in this study. Third, more detailed measures of ND were not available in NHANES. Though TTFC is the best single-item indicator of overall ND [9], this precluded an examination of whether certain dimensions of ND drove the association with more depression symptoms. Further research using larger sample sizes and more objective measures of depression such as clinical diagnoses is needed to examine whether nicotine dependence is associated with the likelihood of clinical depression.

## Implications for public health

This research finds that nicotine dependence is independently associated with higher depression symptom counts, beyond the risk imposed by smoking *behavior* alone. Including TTFC in screening assessments could help more precisely identify those at risk for depression symptoms. Further, considering the long-term physical as well as mental health benefits of quitting among those with depression, specialized intervention efforts are needed to achieve successful outcomes given the increased difficulty faced by this population in successful cessation.

## Author Contributions

**Conceptualization:** Arielle S. Selya.

**Formal analysis:** Tiffany Bainter.

**Methodology:** Arielle S. Selya, S. Cristina Oancea.

**Supervision:** Arielle S. Selya, S. Cristina Oancea.

**Writing – original draft:** Tiffany Bainter, Arielle S. Selya.

**Writing – review & editing:** Arielle S. Selya, S. Cristina Oancea.

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
