## [Decision Letter · Decision Letter 0]

26 Mar 2020

PONE-D-20-00113

A key indicator of nicotine dependence associated with greater depression symptoms, after accounting for smoking behavior

PLOS ONE

Dear Dr. Selya,

Thank you for submitting your manuscript to PLOS ONE. After careful consideration, we feel that it has merit but does not fully meet PLOS ONE’s publication criteria as it currently stands. Therefore, we invite you to submit a revised version of the manuscript that addresses the points raised during the review process.

We would appreciate receiving your revised manuscript by May 10 2020 11:59PM. To enhance the reproducibility of your results, we recommend that if applicable you deposit your laboratory protocols in protocols.io, where a protocol can be assigned its own identifier (DOI) such that it can be cited independently in the future. For instructions see: http://journals.plos.org/plosone/s/submission-guidelines#loc-laboratory-protocols

We look forward to receiving your revised manuscript.

Kind regards,

Neal Doran

Academic Editor

PLOS ONE

Journal Requirements:

Reviewers' comments:

Reviewer's Responses to Questions

**Comments to the Author**

1. Is the manuscript technically sound, and do the data support the conclusions?

Reviewer #1: Partly

Reviewer #2: Yes

2. Has the statistical analysis been performed appropriately and rigorously? 

Reviewer #1: Yes

Reviewer #2: Yes

3. Have the authors made all data underlying the findings in their manuscript fully available?

Reviewer #1: Yes

Reviewer #2: Yes

4. Is the manuscript presented in an intelligible fashion and written in standard English?

Reviewer #1: Yes

Reviewer #2: Yes

5. Review Comments to the Author

Reviewer #1: I appreciate the opportunity to review this manuscript. Overall this is an interesting paper that examines the relationship between Time To First Cigarette (TTFC) after waking and outcomes of depression symptoms among a nationally-representative sample of current adult smokers in the US. The topic is relevant and methods and results are clearly reported. My major concern regarding this manuscript is the idea that the authors consider TTFC as a psychological construct of tobacco addiction. Existing literature has shown that TTFC could be considered as an indicator of physiological nicotine dependence (Branstetter, Muscat, & Mercincavage, 2020; Muscat, Stellman, Caraballo, & Richie, 2009). I would suggest that the authors revise both conceptualizations and elaborate further their reasons to consider this construct as a psychological factor implied in Nicotine Dependence.

Introduction: the distinction between nicotine dependence and smoking behavior is scarcely elaborated. It is needed a more detailed description of each concept.

In the first paragraph of page 4 (lines 69-70), there is a lack of connection between sentences.

In addition, the justification for the study needs strengthening. Authors provide a brief rationale on page 4 (lines 75-76) but this is superficial.

Methods: When describing the instruments, the authors state that “Current smoking behavior was measured as the average number of cigarettes smoked per day during the past 30 days” (page 7, lines 140-141). Nevertheless in the statistical analyses description, the authors stated that “A second model adjusted for both lifetime and current smoking behaviors (pack-years and number of days smoked over the last 30 days, respectively)” (page 7, lines 158-159), and then in Tables, the authors include Cigarettes per Day and Smoking Duration. Please, clarify.

Results: A general description of demographic data of the total sample (N=2070) would have been informative of the sample characteristics (i.e., marital status, employment, health conditions).

Discussion: As mentioned before, in the discussion section it is stated that TTFC is considered as a psychological construct assessing tobacco addiction (page 12, lines 229-230). Nevertheless, literature has shown that TTFC may reflect sensitivity to overnight depletion of blood and brain nicotine concentrations, and therefore TTFC could be considered as an indicator of physiological nicotine dependence (Baker et al., 2007; Fu et al., 2011; Muscat et al., 2009; Walton, Newcombe, Li, Tu, & DiFranza, 2016). In fact, TTFC is one of two items of the Fagerstrom Test for Nicotine Dependence (FTND) which is considered as a quantifiable and objective variable as the number of cigarettes smoked per day. In addition, in the third paragraph (page 12), the authors discuss different reasons for the pattern of results found, focusing on the physiological effects of nicotine instead of psychological ones. I would suggest a general revision of this section in order to clarify and elaborate further on these questions.

Lastly, along the discussion section, the authors allude to smokers with mental illness, smokers with major depression, or smokers with depression despite the fact that they did not find a relation between TTFC with the DSM-5 clinical criteria of Major Depressive Disorder. It would be recommended that the authors made clearer this question, and discuss the pattern of results found referring to depressive symptomatology instead of mental illness or depression.

References

Baker, T., Piper, M., McCarthy, D., Bolt, D., Smith, S., Kim, S. Y., … Toll, B. (2007). Time to first cigarette in the morning as an index of ability to quit smoking: Implications for nicotine dependence. Nicotine and Tobacco Research, 9(SUPPL. 4), 555–570. https://doi.org/10.1080/14622200701673480

Branstetter, S. A., Muscat, J. E., & Mercincavage, M. (2020). Time to First Cigarette. Journal of Addiction Medicine, 1. https://doi.org/10.1097/adm.0000000000000610

Fu, M., Martinez-Sanchez, J. M., Agudo, A., Pascual, J. A., Borras, J. M., Samet, J. M., & Fernandez, E. (2011). Association Between Time to First Cigarette After Waking Up and Salivary Cotinine Concentration. Nicotine & Tobacco Research, 13(3), 168–172. https://doi.org/10.1093/ntr/ntq232

Muscat, J. E., Stellman, S. D., Caraballo, R. S., & Richie, J. P. (2009). Time to first cigarette after waking predicts cotinine levels. Cancer Epidemiology Biomarkers and Prevention, 18(12), 3415–3420. https://doi.org/10.1158/1055-9965.EPI-09-0737

Walton, D., Newcombe, R., Li, J., Tu, D., & DiFranza, J. R. (2016). Stages of physical dependence in New Zealand smokers: Prevalence and correlates. Addictive Behaviors, 63, 161–164. https://doi.org/10.1016/j.addbeh.2016.07.022

Reviewer #2: The manuscript is well-written and addresses an important topic; namely the co-occurrence of nicotine dependence and depressive symptoms. However, I have noted several concerns with the study rationale and approach.

Abstract

- Authors state that, “the psychological construct of ND may be associated with an additional risk for depression symptoms”

-This statement needs to be revised, as their proxy measure of ND is TTFC – which is a behavioral measure of physiological nicotine dependence.

Introduction

-On page 4, the authors note that longitudinal studies have established MDD as a predictor of ND. Thus, one of their stated objectives is to examine ND as a predictor of depression. However, this is not possible to assert given the cross-sectional nature of the data. The nature of their data does allow them to look for an association between ND and depression. However, I don’t think this really supports their argument for why their study is novel/important.

Methods

-It would appear as though the authors’ use the PHQ as their measure of depressive symptoms. If this is the case, it should be cited.

-The term incident rate ratio is inappropriate given the cross-sectional nature of the data.

Results

-A note should be used to explain the meaning of bolded results in the tables.

-It would be helpful to see the Mean(SD) of the depression scores to help with interpretation of the severity of depressive symptoms experienced by the groups.

Discussion

-The authors state, “The findings in this study confirm previous research characterizing the relationship between depression and nicotine dependence”.

-Please revise to “depressive symptoms”

-Cross-sectional data should be listed as limitation.

-Some attention should be given to the lack of a monotonic effect of TTFC on the depression outcomes. How do the authors interpret that finding?

Major Comments

A major concern is the authors’ determination that CPD and TTFC are conceptually distinct, with TTFC being a measure of nicotine dependence (ND) and CPD a measure of smoking behavior. Both of these measures are considered (by most) to be physiological components of nicotine dependence, and are represented as such in many well-established scales including the Fagerstrom Test for Nicotine Dependence and the Heaviness of Smoking Index. The rationale for why these two measures were separately and subsequently modeled as two entirely different constructs has not been adequately fleshed out by the authors.

The authors also assert that, in addition to physiological measures, psychological indicators are important for establishing nicotine dependence, and this point is well-taken. The Nicotine Dependence Syndrome Scale, for example, uses a more comprehensive definition of dependence that includes items assessing Drive (craving and withdrawal, and subjective compulsion to smoke), priority (preference for smoking over other reinforcers), tolerance (reduced sensitivity to the effects of smoking), continuity (regularity of smoking rate), and stereotypy (invariance of smoking. However, there is a disconnect between the rationale laid out in the introduction and their subsequent choice of a single behavioral measure of physiological ND as their proxy measure. The authors’ argument for the unique predictive utility of ND would be much strengthened if their definition was expanded to include psychological components of ND that were alluded to in the Introduction.

6. PLOS authors have the option to publish the peer review history of their article (what does this mean?). If published, this will include your full peer review and any attached files.

Reviewer #1: No

Reviewer #2: No

---

## [Author Response · Author response to Decision Letter 0]

21 Apr 2020

Dear Dr. Doran,

Attached please find our revised manuscript entitled “A key indicator of nicotine dependence is associated with greater depression symptoms, even after accounting for smoking behavior” (PONE-D-20-00113). Thank you for the opportunity to revise our manuscript; we greatly appreciate the feedback of the reviewers. Listed below are the points raised by the reviewers along with our responses. 

Reviewer 1: 

1. My major concern regarding this manuscript is the idea that the authors consider TTFC as a psychological construct of tobacco addiction. Existing literature has shown that TTFC could be considered as an indicator of physiological nicotine dependence (Branstetter, Muscat, & Mercincavage, 2020; Muscat, Stellman, Caraballo, & Richie, 2009). I would suggest that the authors revise both conceptualizations and elaborate further their reasons to consider this construct as a psychological factor implied in Nicotine Dependence.

Our response: Thank you for raising this issue. Our initial draft was written with the assumption that most people outside the subfield of nicotine dependence psychometric research are not aware that there is a distinction between smoking behavior and nicotine dependence (i.e. in our experience, many consider the two terms synonymous). Thus our wording was intended to describe the difference between objective smoking behavior and dependence in a lay fashion, rather than to make the finer-grained distinction between physiological and psychological symptoms. However, we agree now that the term ‘psychological’ is confusing in this respect.

We addressed this by changing our high-level description of ND to a “latent construct” throughout (Abstract, Introduction, and Conclusion). We also cite the two studies you suggested (thank you for pointing these out) in the Introduction as they help to establish the difference between ND and smoking behavior.

2. Introduction: the distinction between nicotine dependence and smoking behavior is scarcely elaborated. It is needed a more detailed description of each concept.

Our response: We have added extensively to the Introduction to discuss objective smoking behavior (cigarettes per day, pack-years), nicotine dependence (listing specific physiological and/or psychological dimensions), and the role of TTFC in particular as an apparently objective behavior that serves as a robust single-item indicator of ND (pp. 3-5). 

3. In the first paragraph of page 4 (lines 69-70), there is a lack of connection between sentences.

Our response: We have added the transition phrase “in examining the relation to depression” so the paragraph now reads: “Several studies have made this important distinction between ND and smoking behavior in examining the relation to depression. Those with depression have higher levels of nicotine dependence…” (p. 5, line 101).

4. In addition, the justification for the study needs strengthening. Authors provide a brief rationale on page 4 (lines 75-76) but this is superficial.

Our response: We have clarified the motivation in this paragraph, highlighting that most existing research did not control for smoking behavior when looking at the relationship between ND and depression, therefore the unique contributions of each are unclear. In the one study to our knowledge that did control for smoking behavior, this was in a Canadian population (while we are looking at a nationally representative US sample) and they controlled for a limited measure of smoking behavior (a binary cutoff of 20 cigarettes/day) while we are controlling for more detailed measures of smoking behavior. The new paragraph reads (p. 5, starting on line 100):

“Several studies have made this important distinction between ND and smoking behavior in examining the relation to depression. Findings such as: those with depression have higher levels of nicotine dependence [4, 20-25], and vice versa [20, 26]. However, very few studies controlled for smoking behavior when examining the association between depression ND [20, 4, 23-25], meaning that it remains unclear what the unique contributions of ND and smoking behavior are to explaining depression outcomes. Longitudinal research has found that major depression is a predictor of earlier TTFC, controlling for a limited measure of current smoking behavior [22] among Canadian adult smokers. Very little is known about the unique contribution of ND in explaining depression, over and above detailed measures of smoking behavior (including smoking history), among a nationally-representative US sample. If ND proves to explain depression more strongly than objective smoking behavior, this is highly relevant for depression screenings among smokers. Previous research has established precedence that ND is a better predictor of other outcomes (e.g. cessation outcomes and smoking biomarkers) when compared to the fairly weak explanatory power of cigarettes per day [11, 26].”

5. Methods: When describing the instruments, the authors state that “Current smoking behavior was measured as the average number of cigarettes smoked per day during the past 30 days” (page 7, lines 140-141). Nevertheless in the statistical analyses description, the authors stated that “A second model adjusted for both lifetime and current smoking behaviors (pack-years and number of days smoked over the last 30 days, respectively)” (page 7, lines 158-159), and then in Tables, the authors include Cigarettes per Day and Smoking Duration. Please, clarify.

Our response: Thank you for catching this error. We have corrected this to “…both lifetime and current smoking behaviors (years of smoking duration and cigarettes per day)” (p. 9, line 193).

6. Discussion: As mentioned before, in the discussion section it is stated that TTFC is considered as a psychological construct assessing tobacco addiction (page 12, lines 229-230). Nevertheless, literature has shown that TTFC may reflect sensitivity to overnight depletion of blood and brain nicotine concentrations, and therefore TTFC could be considered as an indicator of physiological nicotine dependence (Baker et al., 2007; Fu et al., 2011; Muscat et al., 2009; Walton, Newcombe, Li, Tu, & DiFranza, 2016). In fact, TTFC is one of two items of the Fagerstrom Test for Nicotine Dependence (FTND) which is considered as a quantifiable and objective variable as the number of cigarettes smoked per day. In addition, in the third paragraph (page 12), the authors discuss different reasons for the pattern of results found, focusing on the physiological effects of nicotine instead of psychological ones. I would suggest a general revision of this section in order to clarify and elaborate further on these questions.

Our response: Please see our response to your comment #1 above; we now describe ND as a latent construct rather than a psychological construct, which better reflects our intentions. Regarding the nature of TTFC, please also see the additions we made to the Introduction in response to your comment #2 above. Thank you for these references; we have incorporated them into this paragraph in support of TTFC being a physiological indicator of ND (p. 13).

7. Lastly, along the discussion section, the authors allude to smokers with mental illness, smokers with major depression, or smokers with depression despite the fact that they did not find a relation between TTFC with the DSM-5 clinical criteria of Major Depressive Disorder. It would be recommended that the authors made clearer this question, and discuss the pattern of results found referring to depressive symptomatology instead of mental illness or depression.

Our response: We have thoroughly clarified this throughout the Discussion, making it clear that our findings are in relation to the number of depression symptoms rather than a clinical diagnosis. Any remaining mention of “depression” is in relation to other studies we cite.

Reviewer #2:

8. Abstract: Authors state that, “the psychological construct of ND may be associated with an additional risk for depression symptoms.” This statement needs to be revised, as their proxy measure of ND is TTFC – which is a behavioral measure of physiological nicotine dependence.

Our response: Please see our response to Reviewer #1’s comment #1.

9. Introduction: On page 4, the authors note that longitudinal studies have established MDD as a predictor of ND. Thus, one of their stated objectives is to examine ND as a predictor of depression. However, this is not possible to assert given the cross-sectional nature of the data. The nature of their data does allow them to look for an association between ND and depression. However, I don’t think this really supports their argument for why their study is novel/important.

Our response: Regarding the motivation for our study, please see our response to Reviewer #1’s comment #4. Regarding the cross-sectional nature of the data limiting our findings to finding an association, we have revised the wording throughout (“association” and “explaining depression” rather than predicting depression, etc).

10. Methods: It would appear as though the authors’ use the PHQ as their measure of depressive symptoms. If this is the case, it should be cited.

Our response: You are correct; we have now cited the PHQ in Methods/Measures (p. 7, line 146).

11. The term incident rate ratio is inappropriate given the cross-sectional nature of the data.

Our response: “Incident rate ratio (IRR)” has been updated to “prevalence rate ratio (PRR)” throughout.

12. Results: A note should be used to explain the meaning of bolded results in the tables.

Our response: We note in the footnote of Table 2: “bold: p<.05.” (Table 1 contains no boldface cells.)

13. Results: It would be helpful to see the Mean(SD) of the depression scores to help with interpretation of the severity of depressive symptoms experienced by the groups.

Our response: We do present median and interquartile range (IQR) in Table 1 (2nd subrow under the Depression row). Due to the skewed nature of this variable, we chose to present this as median and IQR rather than mean and SD.

14. Discussion: The authors state, “The findings in this study confirm previous research characterizing the relationship between depression and nicotine dependence”.

-Please revise to “depressive symptoms”

Our response: We have changed ‘depression’ has been updated to ‘depressive symptoms (p. 12, line 241).

15. Discussion: Cross-sectional data should be listed as limitation.

Our response: We have now added this to the limitations (p. 15, lines 304-306).

16. Discussion: Some attention should be given to the lack of a monotonic effect of TTFC on the depression outcomes. How do the authors interpret that finding?

Our response: This is an excellent suggestion. We have added the following paragraph to Discussion (p. 13-14, lines 270-277):

“Our findings showed a non-monotonic relationship between TTFC and the number of depression symptoms, such that only those who smoke within 5 minutes of waking are different from those who smoke more than 1 hour after waking. The reasons for this are unclear, but one possibility is a threshold effect such that the association with depression symptoms only manifests at a certain level of ND. Threshold effects have been reported in other related research, e.g. the relationship between TTFC and cotinine levels [REF ]. Further research is needed to examine whether there is a continuous effect of TTFC on depression symptoms, or whether this represents a threshold effect.”

17. A major concern is the authors’ determination that CPD and TTFC are conceptually distinct, with TTFC being a measure of nicotine dependence (ND) and CPD a measure of smoking behavior. Both of these measures are considered (by most) to be physiological components of nicotine dependence, and are represented as such in many well-established scales including the Fagerstrom Test for Nicotine Dependence and the Heaviness of Smoking Index. The rationale for why these two measures were separately and subsequently modeled as two entirely different constructs has not been adequately fleshed out by the authors.

Our response: Thank you for raising this point. We have added extensively to the Introduction (see our response to Reviewer #1’s comment #2) on the distinction between ND and smoking behavior, and explicitly discuss the role of TTFC here.

18. The authors also assert that, in addition to physiological measures, psychological indicators are important for establishing nicotine dependence, and this point is well-taken. The Nicotine Dependence Syndrome Scale, for example, uses a more comprehensive definition of dependence that includes items assessing Drive (craving and withdrawal, and subjective compulsion to smoke), priority (preference for smoking over other reinforcers), tolerance (reduced sensitivity to the effects of smoking), continuity (regularity of smoking rate), and stereotypy (invariance of smoking. However, there is a disconnect between the rationale laid out in the introduction and their subsequent choice of a single behavioral measure of physiological ND as their proxy measure. The authors’ argument for the unique predictive utility of ND would be much strengthened if their definition was expanded to include psychological components of ND that were alluded to in the Introduction.

Our response: Unfortunately, NHANES did not assess more detailed measures of ND; it only included TTFC. Thus, we were not able to examine the contribution of particular dimensions of ND. We have added this as a limitation (p. 15, lines 306-311). 

Thank you for your continued consideration of our manuscript.

Sincerely,

Arielle S. Selya, PhD

Assistant Scientist, Behavioral Sciences Group

Director, Data Exchange Core

Sanford Research

2301 East 60th Street North

Sioux Falls, SD 57104

Email: Arielle.Selya@SanfordHealth.org

---

## [Decision Letter · Decision Letter 1]

11 May 2020

A key indicator of nicotine dependence associated with greater depression symptoms, after accounting for smoking behavior

PONE-D-20-00113R1

Dear Dr. Selya,

We are pleased to inform you that your manuscript has been judged scientifically suitable for publication and will be formally accepted for publication once it complies with all outstanding technical requirements.

With kind regards,

Neal Doran

Academic Editor

PLOS ONE

Additional Editor Comments (optional):

Reviewers' comments:

Reviewer's Responses to Questions

**Comments to the Author**

1. If the authors have adequately addressed your comments raised in a previous round of review and you feel that this manuscript is now acceptable for publication, you may indicate that here to bypass the “Comments to the Author” section, enter your conflict of interest statement in the “Confidential to Editor” section, and submit your "Accept" recommendation.

Reviewer #1: All comments have been addressed

Reviewer #2: All comments have been addressed

2. Is the manuscript technically sound, and do the data support the conclusions?

Reviewer #1: Yes

Reviewer #2: (No Response)

3. Has the statistical analysis been performed appropriately and rigorously? 

Reviewer #1: Yes

Reviewer #2: (No Response)

4. Have the authors made all data underlying the findings in their manuscript fully available?

Reviewer #1: Yes

Reviewer #2: (No Response)

5. Is the manuscript presented in an intelligible fashion and written in standard English?

Reviewer #1: Yes

Reviewer #2: (No Response)

6. Review Comments to the Author

Reviewer #1: The Authors have addressed my concerns throught their responses and changes of the manuscrpt. For this reason I recommend the paper for publication.

Reviewer #2: (No Response)

7. PLOS authors have the option to publish the peer review history of their article (what does this mean?). If published, this will include your full peer review and any attached files.

Reviewer #1: No

Reviewer #2: No

---

## [Editor Report · Acceptance letter]

13 May 2020

PONE-D-20-00113R1 

A key indicator of nicotine dependence is associated with greater depression symptoms, after accounting for smoking behavior 

Dear Dr. Selya:

I am pleased to inform you that your manuscript has been deemed suitable for publication in PLOS ONE. Congratulations! Your manuscript is now with our production department. 

With kind regards,

on behalf of

Dr. Neal Doran 

Academic Editor

PLOS ONE